# The Role of LNK (SH2B3) in the Regulation of JAK-STAT Signalling in Haematopoiesis

**DOI:** 10.3390/ph15010024

**Published:** 2021-12-24

**Authors:** Rhiannon Morris, Liesl Butler, Andrew Perkins, Nadia J. Kershaw, Jeffrey J. Babon

**Affiliations:** 1Walter and Eliza Hall Institute of Medical Research, Parkville, VIC 3052, Australia; morris.r@wehi.edu.au (R.M.); Kershaw@wehi.edu.au (N.J.K.); 2Department of Medical Biology, The University of Melbourne, Parkville, VIC 3052, Australia; 3Australian Centre for Blood Diseases, Monash University, Melbourne, VIC 3001, Australia; liesl.butler@monash.edu (L.B.); andrew.perkins@monash.edu (A.P.); 4Alfred Health, Melbourne, VIC 3001, Australia

**Keywords:** SH2B3, JAK-STAT, myeloproliferative neoplasms

## Abstract

LNK is a member of the SH2B family of adaptor proteins and is a non-redundant regulator of cytokine signalling. Cytokines are secreted intercellular messengers that bind to specific receptors on the surface of target cells to activate the Janus Kinase-Signal Transducer and Activator of Transcription (JAK-STAT) signalling pathway. Activation of the JAK-STAT pathway leads to proliferative and often inflammatory effects, and so the amplitude and duration of signalling are tightly controlled. LNK binds phosphotyrosine residues to signalling proteins downstream of cytokines and constrains JAK-STAT signalling. Mutations in LNK have been identified in a range of haematological and inflammatory diseases due to increased signalling following the loss of LNK function. Here, we review the regulation of JAK-STAT signalling via the adaptor protein LNK and discuss the role of LNK in haematological diseases.

## 1. Introduction

The lymphocyte adaptor protein, LNK (also SH2B3) [1], is a member of the SH2 domain-containing adaptor family of proteins, which also comprises APS (SH2B1) and SH2B (SH2B2). This family of proteins modulate signalling downstream of cytokines and growth hormones. Each protein binds a specific set of tyrosine residues to target proteins via their SH2 domain. In general, APS and SH2B are thought to enhance signalling, whereas LNK acts as a negative regulator (Figure 1). The expression of LNK is predominantly in haematopoietic cells; however, studies have also demonstrated LNK expression in endothelial cells [2] and in neuronal tissue [3]. Here, we discuss the role of LNK in haematopoiesis and haematological diseases specifically.

## 2. Domain Architecture and Protein Structure

SH2B, APS and LNK all share similar domain architecture with three functional domains (Figure 1). The N-terminal domain in each is a dimerisation domain of ca. 60 residues, which is thought to form protein–protein interactions and allow homo or heterodimerisation between SH2B family members [4]. However, homodimerisation of LNK has only been demonstrated in artificial models, and it is unclear whether LNK forms dimers in vivo. The central domain is a pleckstrin homology (PH) domain (ca. 120 residues) that is essential for the localisation of the proteins, forming interactions with phosphatidylinositol lipids in the cell membrane [5]. The C-terminal domain is an Src homology 2 (SH2) domain (ca. 100 residues), which recognises and binds target proteins. Each protein also contains serine and tyrosine residues throughout the protein, which may be phosphorylated in vivo [6]. LNK was initially cloned from rat and mouse DNA libraries [1,7]. Later, using the T-cell acute lymphoblastic leukaemia cell line, Jurkat, the full-length human protein, was predicted to be 63 kDa [8]. The deletion or mutation of various functional elements of LNK revealed that the SH2 domain of LNK is essential for protein function, whereas the C-terminal region (which contains a conserved tyrosine residue) is not required. Interestingly, although the PH domain contributes to LNK function, it is not absolutely essential [9]. 

To date, the only structures of LNK are those of the mouse SH2 domain bound with JAK2 and erythropoietin receptor (EPOR) phosphopeptides (Figure 2) (PDB IDs: 7R8W and 7R8X) [10]. These structures provide insight into how LNK recognises its substrates and how it compares structurally to the other two family members. The LNK SH2 domain adopts the typical SH2 domain fold, with three central beta strands flanked by alpha helices. The LNK SH2 domain, such as the SH2B SH2 domain, is monomeric, whereas APS forms a dimer via its α-B helix [11,12]. In addition to the canonical SH2 domain fold, LNK also contains a short helix on the N-terminal boundary, similar to that observed in other SH2 domains, such as the SOCS and STAT proteins. The structures of the LNK SH2 domains are complex, with JAK2 pY813 and EPOR pY454 peptides. These structures, in addition to biochemical data, indicate that the +1 and +3 residues and, to a lesser extent, the +5 residue downstream of the pY confer specificity for the LNK SH2 domain. The prediction of the full-length structure of human LNK by AlphaFold [13] suggests the PH and dimerisation domains of LNK look similar to the structures of those from APS [4] (PDB IDs: 1V5M and 1Q2H, respectively). However, there are large linker regions between each domain which are predicted to be unstructured and, therefore, how the three domains relate to one another in the full-length protein and whether there is any stable interface between them is unclear (Figure 2).

## 3. LNK as an Adaptor Protein

Because LNK lacks intrinsic catalytic activity, its role as an adaptor protein is to mediate interactions between target proteins and regulatory proteins. To date, LNK has been shown to recruit Castias B-cell lymphoma (CBL) family E3 ubiquitin ligases and the BRISC complex [14,15,16]. CBL and CBL-B are members of the family of RING finger E3 ubiquitin ligases and are expressed in haematopoietic cells. The ring finger domain of these proteins recruit E2 ubiquitin-conjugating enzymes and catalyses the transfer of ubiquitin from E2 to the substrate. This ultimately results in proteasomal degradation of the ubiquitinated target protein [14]. The BRCC36 isopeptidase complex (BRISC) comprises several subunits (KIAA0157, BRCC36, BRCC45 and MERIT40) that form a functional deubiquitinating enzyme complex. Thrombopoietin (TPO) stimulation concurrently increases the K63-ubiquitination and phosphorylation of JAK2 [15]. Loss of BRISC activity was associated with an increase in the K63-ubiquitination of JAK2, along with increased stability and activity. These findings highlight the role of LNK recruitment of CBL proteins and the BRISC complex in the regulation of JAK2 activity. 

## 4. The Role of LNK in the Regulation of JAK-STAT Signalling

LNK was initially thought to be a regulator of T-cell signalling; however, mice deficient in LNK have T-cell numbers similar to those of their WT counterparts. Instead, LNK deficient mice display splenomegaly, an increase in B-cells, hyperplasia of megakaryocytes and increased erythrocyte numbers [7,17]. LNK is highly expressed in haematopoietic stem cells (HSCs), and the deletion of LNK from HSCs leads to an increase in cell number and proliferative capacity [18]. Additionally, dose-dependent over-expression of LNK in lymphoid cells leads to impaired B-cell development, highlighting the role of LNK in the regulation of lymphopoiesis [19]. LNK has been shown to play a particularly important role in the negative regulation of the TPO and erythropoietin (EPO) signalling pathways [20,21], and the effect of LNK on HSC self-renewal and proliferation is hypothesised to be due to its ability to inhibit TPO-mediated signalling [22]. LNK can bind directly to JAK2, the primary JAK activated in response to both EPO and TPO, via its SH2 domain [18]. This binding event occurs at a phosphorylated tyrosine between the kinase and pseudokinase domains (pY813). This residue is also present in JAK3 (but not other members of the JAK family) and is a major site of phosphorylation [23], although the outcome of this phosphorylation in vivo is yet to be elucidated. JAK3 is activated downstream of IL-2 family cytokines [24], including IL-7, and the ability of LNK to regulate B cell development is in accordance with these data. Recently, a study of LNK expression in melanoma found that in melanoma cells, interferon (IFN)-STAT1 signalling induced LNK expression. This increase in LNK expression was thought to ameliorate the anti-proliferative activity of IFN [25], highlighting a role for LNK in the regulation of JAK-STAT signalling in solid tissues as well as in haematopoietic cells.

## 5. Other Potential Targets of LNK SH2 Domain

In addition to JAK2 and JAK3, it has been proposed that the LNK SH2 domain interacts with a suite of phosphorylated sites on various signalling proteins, including c-KIT, c-FMS, FMS-like tyrosine kinase 3 (FLT3) and platelet derived growth factor receptor (PDGFR) [26,27,28,29]. In a recent study of the LNK SH2 domain, residues in the intracellular regions of these receptors, along with several tyrosine residues in the erythropoietin receptor (EPOR), thrombopoietin receptor/myeloproliferative leukemia protein (TPOR/MPL), JAK2 and insulin receptor kinase (IRK), were assessed [10]. Moderate-affinity motifs in EPOR, FLT3 and c-KIT were identified; however, despite reports of LNK associating with PDGFR and c-FMS, the binding of the SH2 domain was not observed. It is possible, however, that other domains of LNK may be associated with these receptors, and the same may be true for MPL and other residues in kinases. 

## 6. Regulation of LNK

The 14-3-3 proteins have been shown to interact with and modulate LNK function [30]. The 14-3-3 proteins form interactions with phosphoserine and phosphothreonine residues on other proteins. Seven 14-3-3 family members have been identified in mammals. The 14-3-3 proteins were shown to directly bind LNK to phosphoserine residues. The binding of 14-3-3 to LNK prevents LNK binding to JAK2 through sequestration, which, in turn, increases JAK2 activity and STAT5 activation downstream of TPO.

## 7. LNK Mutations in Disease

Mutations in the SH2B3, the gene that encodes for LNK, have been identified in patients with a range of diseases, including blood cancers, autoimmune disorders and heart disease [31,32,33,34,35,36,37,38,39,40]. The SH2B3 gene is found on chromosome 12q24.12, and the majority of mutations identified in this gene are missense mutations [41]. Most of these mutations occur within the PH and SH2 domains of LNK, which can lead to mis-localisation or decreased capacity to bind phosphotyrosine, both of which will affect the ability of LNK to regulate signalling. Although mutations in the PH domain appear to be more common than in the SH2 domain, mutations in the SH2 domain have been reported to cause more severe disruption to LNK function [18]. Whether this translates clinically remains to be determined. 

## 8. LNK Mutations in Haematological Cancers

Both germline and somatic LNK mutations have been identified in a range of haematological diseases, including myeloproliferative neoplasms (MPN), myelodysplastic syndromes (MDS), MDS/MPN overlap syndromes and acute lymphoblastic leukaemia (both B and T lineage) [41,42,43,44,45,46,47]. In one study, 5.8% of patients with Philadelphia-like B-ALL had mutations in the SH2B3 [47]. Mutations identified in patients with acute lymphocytic leukaemia have been identified in the PH domain of LNK and caused an increase in cellular proliferation; studies of the mutations in mice showed that the loss of LNK accelerated NOTCH-induced leukaemia [43]. Mice with deletion of LNK and TP53 displayed a pro-B progenitor population with high sensitivity to IL-7 and increased self-renewal capacity, which initiated B-ALL in transplant recipient mice [48]. Further studies identified patients with IL7R-high, LNK-low, high-risk B-ALL associated with Ikaros dysfunction, implicating IL7R/LNK/Ikaros in the oncogenesis of high-risk leukaemia [45]. More recently, somatic mutations in LNK were linked with clonal haematopoiesis, which increases the risk of blood cancer [49]. These studies highlight the role of LNK as a tumour suppressor.

The myeloproliferative neoplasms are a group of diseases characterised by the clonal expansion of mature blood lineages. The Philadelphia negative MPNs comprise essential thrombocythemia (ET), polycythaemia vera (PV) and primary myelofibrosis (PMF). Polycythaemia vera is characterised by the hyperproliferation of erythrocytes [50]. Essential thrombocythemia is characterised by an excess of mature megakaryocytic cells [51] and primary myelofibrosis by the expansion of blood cells in the bone marrow, leading to the formation of scar tissue [52]. The causative mutations of the Philadelphia chromosome-negative MPNs have been well characterised. Most (~97%) cases of PV and 50–60% of cases of ET and PMF are due to the JAK2 p.Val617Phe driver mutation [53,54,55,56,57]. Another 20–25% of ET and PMF are due to driver mutations in exon 9 of calreticulin (CALR) or gain-of-function mutations in MPL. The remaining 10–15% of ‘triple negative’ cases have mutations in many different genes, including SH2B3/LNK. LNK mutations have been reported in up to 7% of MPN cases [31,33,35,58]. LNK mutations can be found in isolation or in conjunction with JAK2, MPL, CALR or other mutations in MPNs and have been described in all types of MPNs. Interestingly, the frequency of LNK mutations increases up to 13% in patients with blast phase transformation, and it is therefore suggested that mutations in LNK may contribute to disease progression. [33]. LNK mutations have also been associated with idiopathic erythrocytosis (IE), an isolated and often inherited expansion of red blood cells [34,59]. Maslah et al., 2017 [41], provide a detailed list of mutations in LNK identified in MPNs and IE. Recently, it was shown using recombinant LNK that mutations within the SH2 domain of LNK identified in patients with either an MPN or IE displayed compromised binding to phosphotyrosine residues that was compromised. Additionally, the expression of full-length LNK in cells revealed that these mutations had a decreased capacity to regulate signalling when compared with the WT protein [10]. In addition to regulating normal haematopoiesis, WT LNK is a potent negative regulator of mutant JAK2 and MPL in vitro [60,61], inhibiting EPO and TPO signalling in cells transformed with mutant alleles. LNK binds JAK2 V617F and MPL W515L and inhibits the activation of JAK2, STAT3, ERK and AKT, constraining proliferation. As such, the disruption of LNK function with increased signalling driven by JAK2, MPL and CALR mutations in part explains why a concurrent mutation in these proteins may lead to more severe disease in MPNs [33].

## 9. Mutations Identified in the LNK SH2 Domain

Given that the SH2 domain of LNK is indispensable for activity, mutations affecting its function are of interest. Several mutations within the LNK SH2 domain (Figure 3) have been associated with an increased proliferative capacity of haematopoietic cells. The majority of these mutations were identified in patients with MPNs or the related IE (W364X, S370C, L390W, R392Q, S394C, E395K, E400K, V402M, R415H, R415C, R425C, M437I and I446V) [36,41,59,62,63,64,65,66,67]; however, several were also identified in patients with B-cell precursor ALL (R392W, R397G and Q427P) [43,68,69]. These mutations tend to cluster in areas important for peptide binding, particularly around the phosphotyrosine binding pocket (Figure 3), and so may impede the association of LNK with target proteins, as has been shown for V402M and R415C/H [10]. Mutations not involved directly in binding may still alter the association of the LNK SH2 domain with targets by reducing the stability of the SH2 domain, reducing affinity for binding partners by altering the position of key residues or, in the case of the W364X mutation, by introducing a premature stop codon, thereby eliminating the SH2 domain entirely. 

## 10. LNK as a Potential Therapeutic Target

Given the role of LNK in normal and malignant haematopoiesis, there is interest in targeting LNK with small molecules or enriching LNK expression to treat disease. The role of LNK in regulating haematopoietic stem cell quiescence suggests it could be a target for mobilising HSCs or for increasing progenitor expansion following a bone marrow transplant [9]. In mice, the deletion of LNK was shown to restore HSC function in Fancd2 deficient Fanconi anaemia [70], highlighting the effect that the downregulation of LNK can have on this cell population. Given that the LNK SH2 domain has been shown to be indispensable, this domain could be targeted with a small molecule to block association with target proteins. The expression of LNK in leukemic cells was shown to suppress their proliferation after the delivery of LNK with an octa-arginine peptide inhibited the proliferation of M-MOK leukaemia cells [71]. These findings support the idea that increasing LNK expression can inhibit the cellular proliferation of leukemic cells. However, gene therapy is a challenging task, and it is unclear whether there is another way to enhance LNK expression in specific cells, particularly without side effects such as thrombocytopenia.

## 11. Unanswered Questions and Conclusions

One of the main uncertainties that remain is how LNK exerts its negative regulatory functions. Recent work has demonstrated that LNK recruits CBL, CBL-B and the BRISC complex, which control JAK2 stability and degradation [14,15]. However, whether other mechanisms also exist remains to be demonstrated. LNK may also be able to change the conformation of kinases such as the JAKs and keep them in an inactive state. Additionally, the binding of LNK to phosphotyrosine residues on receptors may prevent other proteins from binding and activating downstream pathways, with LNK acting as a competitive inhibitor. In addition to how LNK exerts its activity, it is unclear why, given their similar structures, APS and SH2B appear to upregulate signalling, while LNK is a negative regulator. This may be explained by the ability of each protein to recruit other proteins that have catalytic activity that ubiquitinate, dephosphorylate or regulate kinase activity or by their ability to dimerise and bring kinases closer together for transphosphorylation and activation. Another outstanding uncertainty is how LNK exerts specificity for different signalling pathways. Given that the LNK SH2 domain is associated with JAK2 and JAK3, which are involved in signalling downstream of numerous cytokines, it is unclear why LNK appears to only regulate signalling for a subset of these cytokines. It has been suggested that LNK is induced upon signalling via some cytokines [72]; however, the change in LNK expression is minimal in comparison to the SOCS proteins, which are known to be negative feedback inhibitors of cytokine signalling. It is also possible that other domains or regions within LNK outside of the SH2 domain are important for targeting specific pathways in ways we do not yet understand. Additionally, while the role of the SH2 domain of LNK is clear, the functions of the other motifs and domains of LNK are yet to be studied in greater detail. Understanding the structure and function of full-length LNK may also provide some insight into the former questions. Additionally, binding studies using the full-length protein would allow for a more complete picture of the full binding repertoire of LNK.

## Figures and Tables

**Figure 1 pharmaceuticals-15-00024-f001:**
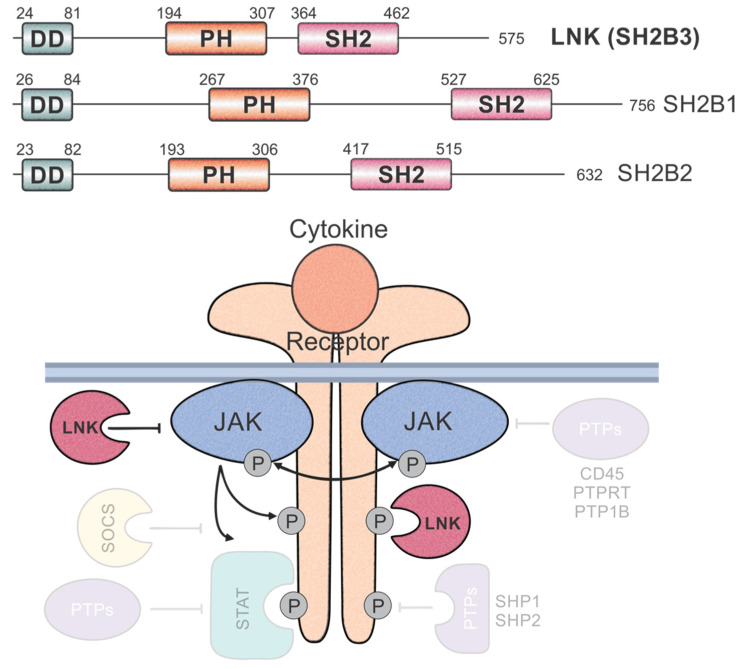
LNK and the adaptor family of proteins. The SH2 domain-containing adaptor family of proteins all share the same domain architecture (**top**) comprising an N-terminal dimerisation domain (DD), central pleckstrin homology domain (PH) and C-terminal Src homology 2 (SH2) domain. Schematic diagram of cytokine signalling showing regulators of cytokine signalling and where they act (**bottom**). Phosphatases and Suppressor of Cytokine Signalling (SOCS) proteins are shown semi-transparently, whereas LNK is shown in solid colour. LNK directly binds JAKs and receptors via its SH2 domain and inhibits downstream signalling.

**Figure 2 pharmaceuticals-15-00024-f002:**
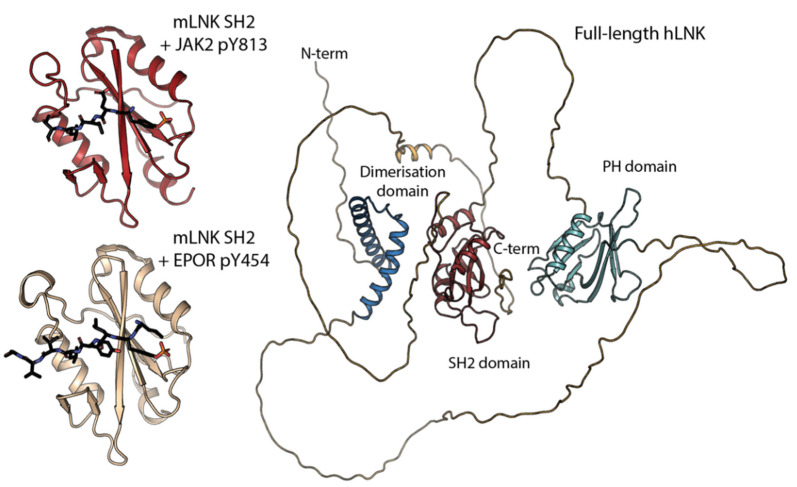
Structure of LNK (SH2B3). The domain architecture of full-length LNK (**right**, prediction from AlphaFold) includes three functional domains. The first is a dimerisation domain, a central pleckstrin homology domain and a C-terminal SH2 domain joined by what is predicted to be unstructured regions (**left**). The crystal structures of the *M. musculus* LNK SH2 domain with JAK2 pY813 (PDB ID: 7R8W) and EPOR pY454 (PDB ID: 7R8X) peptides.

**Figure 3 pharmaceuticals-15-00024-f003:**
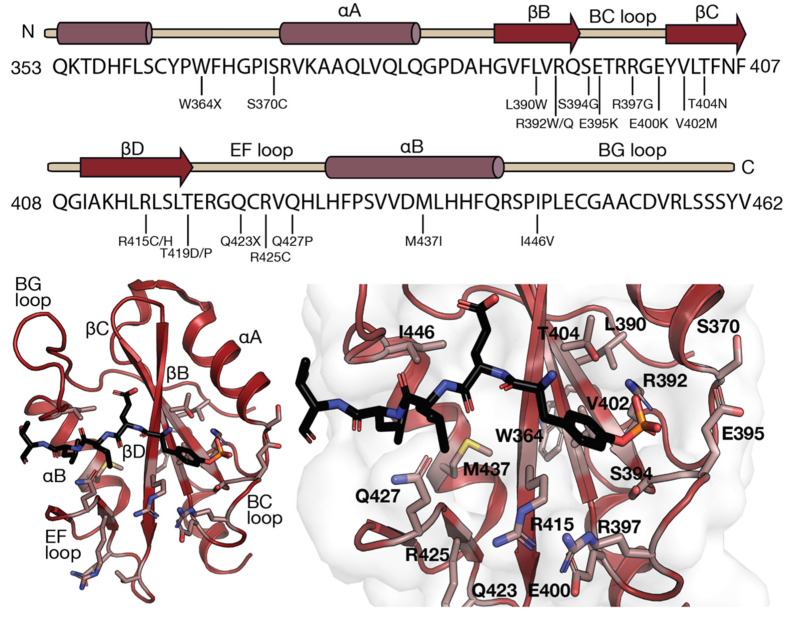
Mutations located within the LNK SH2 domain. Secondary structural elements of the LNK SH2 domain with the position of mutations identified within the SH2 domain indicated (**top**). Structure of the *M. musculus* LNK SH2 domain with the JAK2 pY813 peptide bound (PDB ID: 7R8W). Residues identified as mutation sites are shown as sticks and highlighted in pink (*H. sapiens* LNK SH2 domain numbers are indicated, all residues are conserved in *M. musculus*).

## Data Availability

No new data were created or analyzed in this study. Data sharing is not applicable to this article.

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
