# Peer review of "The Role of LNK (SH2B3) in the Regulation of JAK-STAT Signalling in Haematopoiesis"

_pharmaceuticals, 2021, doi:10.3390/ph15010024_

Round 1

Reviewer 1 Report

Review Morris et al., pharmaceuticals-1502235

In the present study, Morris and colleagues review the role of LNK in JAK-STAT signaling. The manuscript contains information on the current understanding of LNK as a negative regulator attenuating mainly JAK2 and JAK3 downstream signaling. Although the precise mechanism is not completely understood, LNK is thought to have an adaptor-like function for other proteins regulating JAK kinases. The second part of the review deals with LNK loss-of-function mutations that have been discovered in several malignant and non-malignant diseases.

In general, the manuscript requires an intensive revision, structurally as well as stylistic. In addition, the reference need to be updated and some orthographic issues require attention.

Detailed comments:

  1. Please revise the manuscript style. A number of sentences are too long for easy understanding, please shorten/ split them according to the “one sentence, one information” rule. Examples are lines 39-43, 80-83, 96-100, 118-121, 160-165, 168-171, 222-225.
  2. Moreover, some phrases are not up to scientific standards, e.g. 145-147, 135-137, 204-213, 219-220, 227-230, 231-234. Please rewrite.
  3. The manuscript contains a number of typos or grammatical issues, e.g. line 39 (comma missing), line 59 “central”, 165 (comma missing), 182 (“affecting”), 188 (“to” missing), 201 (two periods), 216 (“thrombocytopenia”), 217 (remove period), 225 (“it” missing), 226 (comma missing), 227 (comma missing), 228 (comma missing), 236 (comma missing).
  4. The paragraph that should deal with the role of LNK in JAK-STAT signaling misses information on the actual (adaptor) role of LNK. The part describing expression patterns of LNK should instead be moved at the very beginning of the review, before domain structure is explained.
  5. The structure of the part on LNK mutations is confusing. Why is there a separate paragraph on MPN, as this is a blood cancer? The part on LNK mutations in non-malignant disease, on the other hand, is very superficial.
  6. The explanation of MPN types and mutations is rather long and could be reduced to some short statements, as the connection to LNK mutations is unclear.
  7. In general, the weighting of subtopics does not correspond to the title, as the major part of the manuscript deals with the description of the role of LNK (mutations) in various diseases. Consider elaborating the part where you actually describe the role of LNK in JAK/STAT signaling.
  8. The cited literature is mostly older than 10 years, the review misses some “most relevant” hits retrieved on a Pubmed or GoogleScholar search “LNK”, e.g. Lv et al., Cancer Cell Int 2020, Zhong et al., Aging 2020, Pan et al., Biomed Pharmaother 2020. Please include more recent literature.

Author Response

We sincerely thank reviewer 2 for their kind comments and their excellent queries and feedback on our manuscript.
Please see below for a detailed response to each of the queries.

Please revise the manuscript style. A number of sentences are too long for easy understanding, please shorten/ split them according to the “one sentence, one information” rule. Examples are lines 39-43, 80-83, 96-100, 118-121, 160-165, 168-171, 222-225.

Response: We thank reviewer 1 for this feedback. We have edited each of the sentences mentioned, and have split them into two or more sentences. In addition we have edited other areas of the manuscript with this point in mind.

Moreover, some phrases are not up to scientific standards, e.g. 145-147, 135-137, 204-213, 219-220, 227-230, 231-234. Please rewrite.

Response: We thank reviewer 1 for highlighting these phrases. The sentences and/or paragraphs highlighted have been rephrased or entirely rewritten.

The manuscript contains a number of typos or grammatical issues, e.g. line 39 (comma missing), line 59 “central”, 165 (comma missing), 182 (“affecting”), 188 (“to” missing), 201 (two periods), 216 (“thrombocytopenia”), 217 (remove period), 225 (“it” missing), 226 (comma missing), 227 (comma missing), 228 (comma missing), 236 (comma missing).

Response: We thank reviewer 1 for highlighting these typos and grammatical errors. We have reviewed and edited the errors highlighted above and throughout the manuscript.

The paragraph that should deal with the role of LNK in JAK-STAT signaling misses information on the actual (adaptor) role of LNK. The part describing expression patterns of LNK should instead be moved at the very beginning of the review, before domain structure is explained.

Response: We thank reviewer 1 for this valuable feedback. We have moved the LNK expression patterns into the first paragraph.

In addition we have added a section discussing the role of LNK as an adaptor protein and its regulation as follows: 

"LNK as an adaptor protein

Because LNK lacks intrinsic catalytic activity its role as an adaptor protein is to mediate interactions between target proteins and regulatory proteins. To date, LNK has been shown to recruit Castias B-cell lymphoma (BCL) family E3 ubiquitin ligases and the BRISC complex [14-16]. CBL and CBL-B are members of the family of RING finger E3 ubiquitin ligases and are expressed in haematopoietic cells. The ring finger domain of these proteins recruits E2 ubiquitin conjugating enzymes and catalyse transfer of ubiquitin from the E2 to substrate. This ultimately results in proteosomal degradation of the ubiquitinated protein. The BRCC36 isopeptidase complex (BRISC) comprises several subunits (KIAA0157, BRCC36, BRCC45 and MERIT40) that form a functional deubiquitinating enzyme complex. TPO stimulation concurrently increases K63-polyubiqutinitation and phosphorylation of JAK2. Loss of BRISC activity was associated with an increase in K63 ubiquitination of JAK2 along with increased stability and activity. These findings highlight the role of LNK recruitment of CBL proteins and the BRISC complex in the regulation of JAK2 activity."

"Regulation of LNK

The 14-3-3 proteins have been shown to interact with, and module LNK function [30]. The 14-3-3 proteins form interactions with phosphopersine and phosphothreonine resides on other proteins. Seven 14-3-3 family members have been identified in mammals. The 14-3-3 proteins were shown to directly bind LNK at phosphoserine residues. Binding of 14-3-3 to LNK prevents binding to JAK2 through sequestration, which in turn increases JAK2 activity and STAT5 activation downstream of TPO."

The structure of the part on LNK mutations is confusing. Why is there a separate paragraph on MPN, as this is a blood cancer? The part on LNK mutations in non-malignant disease, on the other hand, is very superficial.

Response: In order to be more concise, we have changed the focus of the review to solely a discussion of LNK in haematological diseases, and as such have removed the section on non malignant disease. The MPN section has now become a section of the blood cancer section as we agree this structure makes more sense. 

The explanation of MPN types and mutations is rather long and could be reduced to some short statements, as the connection to LNK mutations is unclear.

Response: We thank reviewer 1 for this valuable feedback.

We removed the discussion of the two groups of MPNs for clarity, however we felt the discussion of the most 3 common mutations was required as we later discuss that LNK can be mutated in conjunction with JAK2/MPL/CALR,. Additionally we felt that including this information gives context to the low incidence of LNK mutations as drivers of MPNs, and instead its role in leukemic transformation.

In general, the weighting of subtopics does not correspond to the title, as the major part of the manuscript deals with the description of the role of LNK (mutations) in various diseases. Consider elaborating the part where you actually describe the role of LNK in JAK/STAT signaling.

Response: We have edited the title of the paper to reflect more clearly the theme of the review - LNK in the regulation of JAK-STAT in hematopoiesis. With the addition of sections discussing the regulation of LNK, and role as an adaptor protein, the manuscript is now more evenly evenly split between discussion of LNK function in the regulation of JAK-STAT singalling and the role LNK mutations play in disease.

The cited literature is mostly older than 10 years, the review misses some “most relevant” hits retrieved on a Pubmed or GoogleScholar search “LNK”, e.g. Lv et al., Cancer Cell Int 2020, Zhong et al., Aging 2020, Pan et al., Biomed Pharmaother 2020. Please include more recent literature.

Response: We thank reviewer 1 for this valuable feedback.

The above literature mentioned was not cited as these papers do not discuss JAK-STAT signalling in the context of haematology, which we have chosen to change our entire focus to. However, with the addition of the more detailed discussion of the adaptor role of LNK and its regulation, more recent literature has been cited.

Reviewer 2 Report

This is a nice mini review regarding the adaptor protein LNK with a focus on its function and mutations in hematological diseases. Although it is well-written, further proofreading is necessary before publication. For examples, On line 244, "diseases" between "hematological and including" is missing. On line 136, "that" should be added between "suggest and LNK". 

Author Response

We sincerely thank reviewer 1 for their kind comments and feedback. 

We have proofread and edited the manuscript further as per the reviewers suggestion and thank them for highlighting several missing words which have been added or edited.

Round 2

Reviewer 1 Report

The authors have addressed all of my concerns sufficiently.